# A mechanism for reconciling the synchronisation of Heinrich events and Dansgaard-Oeschger cycles

Clemens Schannwell [1] ✉, Uwe Mikolajewicz[1], Marie-Luise Kapsch [1] & Florian Ziemen [2]

The evolution of the northern hemispheric climate during the last glacial period was beset by quasi-episodic iceberg discharge events from the Laurentide ice sheet, known as Heinrich events (HEs). The paleo record places most HEs into the cold stadial of the Dansgaard-Oeschger cycle. However, not every Dansgaard-Oeschger cycle is associated with a HE, revealing a complex interplay between the two modes of glacial variability. Here, using a coupled ice sheet-solid earth model, we introduce a mechanism that explains the synchronicity of HEs and Dansgaard-Oeschger cycles. Unlike earlier studies, our mechanism does not require a trigger during the stadial. Instead, the atmospheric warming signal during the interstadial of the Dansgaard-Oeschger cycle causes enhanced ice stream thickening that leads to the HE during the late interstadial. We demonstrate that this mechanism reproduces the key HE characteristics and provides an explanation for synchronous HEs from different regions of the Laurentide ice sheet.

Heinrich events are associated with massive iceberg discharge events from the Laurentide ice sheet - the ice mass covering the North American continent during past glacial periods. Subsequent melting of the icebergs left behind a distinct layer of ice-rafted debris on the North Atlantic ocean floor that ultimately led to the identification of Heinrich events[1,2]. During the Marine Isotope Stage 3 period (MIS3; ca. 60-25 ka BP), four Heinrich events have been identified in the paleo records[2]. Remarkably, most of them occurred during the cold phase of an Dansgaard-Oeschger cycle[3], a period that is more commonly associated with ice mass gain rather than ice mass loss. Dansgaard-Oeschger cycles are characterised by a period of rapid, decadal warming of up to 14° C in the high northern latitudes[4], followed by a more gradual cooling spanning several centuries. Temperature reconstructions from ice cores indicate a dominant recurrence interval of ~1,500 years for Dansgaard-Oeschger cycles[5]. To this day, the underlying mechanisms interlinking Dansgaard-Oeschger cycles and Heinrich events remain poorly understood.

Over the last decades many trigger mechanisms for Heinrich events have been put forward in search of a better understanding of the governing processes as well as reproducing the characteristics of Heinrich events as derived from the paleo record[6–10]. Early theories focused on internally-driven oscillations[6,7,11–13], suggesting that Heinrich events follow a two-stage pattern referred to as binge-purge cycle[6]. During the binge phase, the ice sheet becomes thicker causing an increase in basal temperatures until it crosses a critical thickness and temperature threshold, resulting in the onset of rapid basal sliding. The ensuing purge phase is characterised by elevated ice discharge that is maintained until the internal heat supply decreases and basal temperatures fall below the pressure-melting point. The validity of this mechanism has since been challenged because it struggles to explain the observed timing of Heinrich events within the Dansgaard-Oeschger cycle[8,9]. The occurrence of most Heinrich events during cold stadials, also referred to as phase locking[7,14], suggests a common external climatic trigger. Motivated by observational evidence indicating an ocean subsurface warming signal prior to several Heinrich events[15,16], more recent theories revolve around the ocean as key climate component for controlling the timing of Heinrich events[8–10]. Some studies suggest that the observed subsurface warming of the

[1]Max Planck Institute for Meteorology, Bundesstraße 53, 20146 Hamburg, Germany. [2]Deutsches Klimarechenzentrum, Bundesstr. 45a, 20146 Hamburg, Germany. ✉e-mail: clemens.schannwell@mpimet.mpg.de

ocean led to the demise of an ice shelf wedged-in between Greenland and the Laurentide ice sheet[8,9]. The sudden disintegration of the ice shelf destabilised the Hudson ice stream, leading to an acceleration of the ice located upstream, very similar to the recently reported behaviour of regions in the Antarctic[17]. While this may explain the timing of Heinrich events within the Dansgaard-Oeschger cycle, it is incompatible with proxy observations, which provide no firm evidence of the existance of such a persistent ice shelf. This shortcoming inspired the development of a mechanism that links Heinrich events to the intrusion of warm subsurface ocean waters, as observed at marine outlet glaciers in Greenland and Antarctica[10]. The warm ocean water initiates widespread ice-sheet retreat through a positive feedback mechanism referred to as marine ice cliff instability. Subsequent glacio-isostatic uplift, as a result of the reduced ice mass, cuts off the path of the warm ocean waters to the ice stream, terminating the Heinrich event and leading to a regrowth of the ice sheet. The use of simple coupled models, however, has revealed that two weakly coupled autonomous oscillators, in this case the ice stream and the ocean, with different natural frequencies can synchronise with ice-ocean interactions alone, if the external forcing that both oscillators are exposed to is strong enough to overcome the natural frequency of the two oscillators[14,18,19].

All previous theories have in common that they have been specifically tailored towards the glaciological and climatological setting of the Hudson ice stream[7–10,19]. Some studies[7,10] have hinted, but without demonstrating it, that their mechanism could explain the observational evidence of synchronous Heinrich events from Hudson ice stream and other regions of the Laurentide ice sheet[20]. Naturally, synchronous Heinrich events from mechanisms based on the permanently marine-terminating Hudson ice stream are restricted to ice streams with such characteristics. However, the proxy record cannot dismiss the idea of synchronous Heinrich events from land-terminating ice streams that only become marine-terminating during the actual Heinrich event[21]. Support for this is provided by mapped geomorphological features from land-terminating ice streams of the Laurentide ice sheet that indicate surging[22,23], even though the timing of these surges remains poorly constrained. A modern-day analogue for this hypothesis, albeit on a much smaller spatial scale than Heinrich events, comes from observations of synchronous surges from land-terminating mountain glaciers[24]. In the following, we propose a mechanism for Heinrich events that can trigger a pan-ice sheet response and is applicable to land and marine-terminating ice streams exhibiting distinctly different climatological settings. As a result, we are able to explain all of the dominant observed characteristics from the paleo record.

## Results and discussion
### Phase locked Heinrich events from the Hudson ice stream
Heinrich events in our simulations are preconditioned by the binge-purge mechanism[6]. However, the final trigger for the Heinrich events that determines the exact onset of the purge phase is provided by the Dansgaard-Oeschger cycle forcing. Conceptually, in our mechanism the rapid transition from stadial to interstadial results in a surface temperature warming that is accompanied by an elevated surface mass balance. Changes in surface temperature and surface mass balance instantaneously affect the geometric and thermal evolution of the ice stream. The warmer surface temperatures lead to increased ice deformation and enhanced frictional heat generation, while the higher surface mass balance flux increases ice stream thickening. Together these processes ensure that a critical ice thickness and basal temperature threshold, required for the onset of the internal instability, is reached towards the end of the interstadial phase (Fig. 1a, b). Because a Heinrich event is only triggered when the size of the region permitting rapid basal sliding crosses a certain threshold, the onset of the purge phase is typically delayed into the late interstadial of the Dansgaard-Oeschger cycle (Fig. 1c). Our canonical simulated Heinrich event lasts between 1000 and 1200 years and adds between 2.7 and 4 m to the

global sea level with peak discharge rates reaching up to 45 mSv. In accordance with the paleo record, not every Dansgaard-Oeschger cycle is associated with a Heinrich event[3], but, owing to the ice stream's natural frequency, it takes between 3-5 Dansgaard-Oeschger cycles for the next Heinrich event to occur.

To demonstrate our mechanism we use simulations with a coupled ice sheet-solid earth model. Under constant MIS3 climate forcing, Heinrich events from the Hudson ice stream do not exhibit an inherent phase locking signal to Dansgaard-Oeschger cycles (henceforth referred to as reference simulation; see Methods: Initial state and Dansgaard-Oeschger cycle forcing). However, if we prescribe a 1500 year long Dansgaard-Oeschger cycle in addition to the constant MIS3 climate forcing, Heinrich events cluster in the late interstadial phase during a Dansgaard-Oeschger cycle (Fig. 2). This is illustrated by comparing an ensemble of simulations with prescribed and spatially homogeneous Dansgaard-Oeschger cycle forcings with the reference simulation (Supplementary Fig. 1). For the construction of our ensemble simulations mimicking a Dansgaard-Oeschger cycle forcing, we focus on two key parameters: (i) the surface temperature and (ii) the surface mass balance, which have previously been shown to affect the timing of Heinrich events[25,26]. The prescribed magnitudes range from 2 to 14° C for surface temperature ($\Delta$Temp, see Methods: Dansgaard-Oeschger cycle forcing) and 0.009–0.1 m yr$^{-1}$ for the surface mass balance ($\Delta$SMB) and are motivated by reported values from ice core reconstructions[4] as well as model simulations[27]. When we compare the empirical Heinrich event distributions of the forced simulations with the unforced reference simulation, we find that, based on the Kuiper's test for two distinct empirical distributions (see Methods: Coupled simulations), 17 out of the 24 forced simulations show a Heinrich event distribution that is statistically significantly ($R^2 > 0.9$) different from the reference simulation and exhibit phase locking (Supplementary Fig. 2). Only the smaller temperature and surface mass balance simulations do not show phase locking. The strength of the observed phase locking becomes stronger for larger surface mass balance perturbations which is in accordance with an earlier study[26]. Additional sensitivity simulations reveal that for the stronger perturbation scenarios ($\Delta$Temp $\geq$ 10 °C and $\Delta$SMB $\geq$ 0.075 m yr$^{-1}$) only one of the two atmospheric forcings is required to achieve phase locking (Supplementary Fig. 3). Hereby, perturbations to only the surface mass balance result in Heinrich events occuring deeper into the stadial phase of the Dansgaard-Oeschger cycle. This underlines that the timing of internally-driven Heinrich events is modulated by the prescribed atmospheric perturbations.

Since the paleo record indicates that Heinrich stadials tend to be longer than non-Heinrich stadials[4,28], we performed a subset of the ensemble with a shorter 1000-year-long Dansgaard-Oeschger cycle, a 2000-year-long Dansgaard-Oeschger cycle, and experiments in which the Dansgaard-Oeschger cycle varies stochastically between 1000 and 2000 years in 100 year increments. Our simulations show that longer Dansgaard-Oeschger cycles result in enhanced phase locking and push the onset of the Heinrich events deeper into the stadial of the Dansgaard-Oeschger cycle (~135 years later, Supplementary Fig. 4). In contrast, the shorter 1000 year long Dansgaard-Oeschger cycle shows weaker phase locking (Supplementary Fig. 4). The response to an enhanced noise level in the Dansgaard-Oeschger cycle length is that the phase locking becomes weakened (Supplementary Fig. 5). Despite the Heinrich events being more spread out, the vast majority of the events still occurs during the cooling phase of the Dansgaard-Oeschger cycle (Supplementary Fig. 5). Many events show even an onset much deeper into the stadial phase which is more in accordance with reconstructions. Regardless of the length of the Dansgaard-Oeschger cycle, phase locking is persistent in almost all our simulations. This is corroborated by additional simulations with a more realistic, spatially-varying forcing (Supplementary Fig. 6). These simulations also display a distinct phase locking signal (Supplementary Fig. 7).

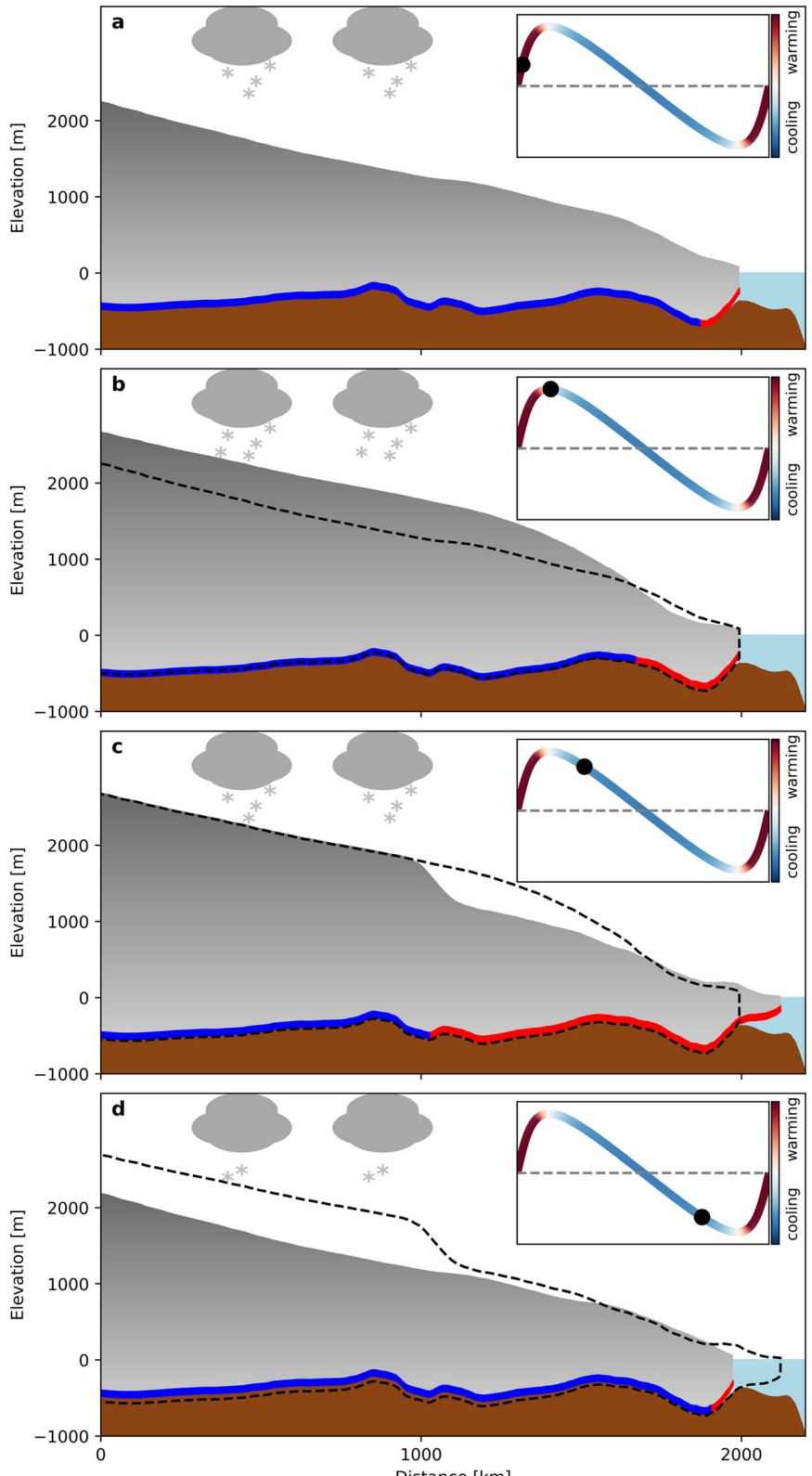

**Fig. 1 | Stages of the proposed Heinrich event mechanism. a** Enhanced build-up phase of the ice stream during interstadial. **b** Ice stream reaches critical ice thickness and temperature threshold for an Heinrich event during the inter-stadial. **c** Onset of Heinrich event during the late interstadial. **d** Shutdown of the Heinrich event at the end of the stadial. Dashed outline shows position from previous snapshot. Red colour at the bottom of the ice stream shows regions where ice is at the pressure-melting point while the blue colour shows regions where ice is frozen to the ground. Insets are created using Eq. (3) from the "Methods" section. Colours represent the time derivative of the temperature and surface mass balance forcing.

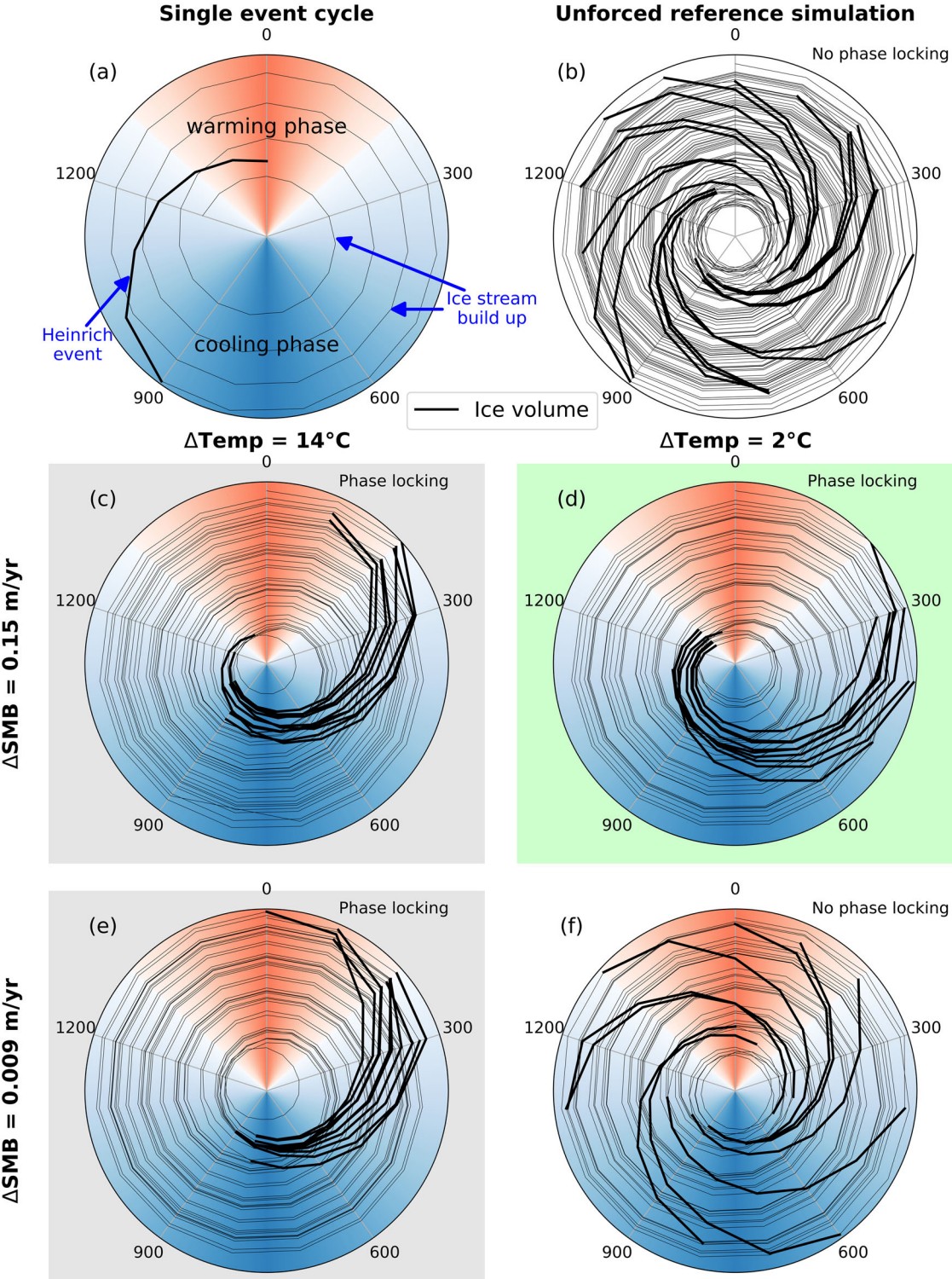

**Fig. 2 | Phase locking plot for Hudson ice stream. a** Ice volume evolution for the Hudson ice stream for a single Heinrich event cycle, (**b**) the unforced reference simulation and (**c**–**f**) the endmembers of the ensemble with prescribed Dansgaard-Oeschger cycle forcings. The polar plot relates ice volume on the radial axis to the Dansgaard-Oeschger cycle on the circle's circumference. Red and blue shadings in each circle correspond to the warming and cooling phase of the Dansgaard-Oeschger cycle. Heinrich events are highlighted as thick black lines. Phase locking

occurs when individual Heinrich events show a clustering. A denser clustering hereby represents stronger phase locking. In the extreme case of perfect phase locking, all Heinrich events would plot on top of each other and reduce to a single line. Green, grey and white background colours mark simulations with Heinrich event distributions that are different from the reference simulation at the 1%, between 1–10%, and >10% significance level, respectively. Radial ice volume axes range from $4 \times 10^{15}$ – $5.9 \times 10^{15}$m³, covering ~4.8 metres of sea-level equivalent.

## Synchronous Heinrich events from the Laurentide ice sheet

The main characteristic of Heinrich events from the paleo record that previously proposed mechanisms[7–10] have failed to demonstrate is the phenomenom of synchronous Heinrich events from the Hudson ice stream and other regions of the Laurentide ice sheet. Paleo ice stream reconstructions indicate that other ice streams may have been active during a number of the recorded Heinrich events[20], such as Mackenzie ice stream, which is located in the north-west of the Laurentide ice sheet. Here, we show that our proposed Heinrich mechanism also leads to phase locking for Mackenzie ice stream, albeit weaker than for Hudson ice stream, resulting in synchronous Heinrich events from both ice streams. Mackenzie ice stream is located in a distinctly different glaciological and climatological setting in comparison to Hudson ice stream, but also exhibits Heinrich-like events in our simulations. Therefore, it presents an ideal case to test the generalisation of our mechanism. The surface temperatures are ~8–10 °C warmer than in the Hudson area, resulting in shorter Heinrich event intervals. The timing of the events is primarily controlled by surface temperature rather than surface mass balance[26]. During the binge phase, Mackenzie ice stream terminates on land and only becomes marine-terminating for short periods during the purge phase. Due to the absence of a well-confined subglacial trough, the periodicity of Heinrich events for Mackenzie ice stream shows an enhanced variability. These circumstances result in a noisier distribution of Heinrich events with weaker phase locking throughout our experiments in comparison to Hudson ice stream (Supplementary Fig. 8). However, phase locking becomes stronger for larger surface temperature perturbations. Similar to the Hudson ice stream, longer Dansgaard-Oeschger cycles also result in stronger phase locking and push the occurence of Heinrich events into the stadial phase of the Dansgaard-Oeschger cycle (Supplementary Fig. 9).

To illustrate that our proposed mechanism supports synchronous Heinrich events from Hudson and Mackenzie ice streams, we use our simulation ensemble with spatially homogeneous Dansgaard-Oeschger perturbations and compute the likelihood of a Heinrich event from Mackenzie ice stream within a time window of 300 and 600 years of a Heinrich event from Hudson ice stream. To improve the signal-to-noise-ratio, we average the likelihood of synchronous Heinrich events across the surface mass balance ranges. This step is justified because the periodicity of Heinrich events from Mackenzie ice stream has shown to be more sensitive to surface temperature changes[26] than surface mass balance changes[26]. The results show that the likelihood of synchronous Heinrich events substantially increases for higher surface temperature perturbations from ~8% to ~17% and ~8% to ~22% for the 300 year time window and for the 600 year window, respectively (Fig. 3). This corresponds to a total of four more synchronous Heinrich events for the 300 year window and six more synchronous Heinrich events for the 600 year window. An even stronger signal for synchronous Heinrich events (up to 50%) is observed for the longer Dansgaard-Oeschger cycle experiments, supporting the result that longer Dansgaard-Oeschger cycles enhance phase locking. As a consequence, every fifth Heinrich event for the 1500-year-long Dansgaard-Oeschger cycle and every second Heinrich event for the 2000-year-long Dansgaard-Oeschger cycle from the Hudson ice stream is accompanied by a Heinrich event from the Mackenzie ice stream. This matches well with reconstructions from ice-rafted debris that show contributions from source regions other than the Hudson area for some, but not all recorded Heinrich events[20,29].

Here, we have introduced a Heinrich event mechanism in which internally driven ice-sheet oscillations are paced by an atmospheric perturbation signal caused by Dansgaard-Oeschger cycles. Unlike earlier theories, our mechanism does not need a trigger event during the stadial phase of the Dansgaard-Oeschger cycle. This mechanism successfully reproduces all of the main characteristics of Heinrich events as evidenced by the paleo record. We demonstrate that our

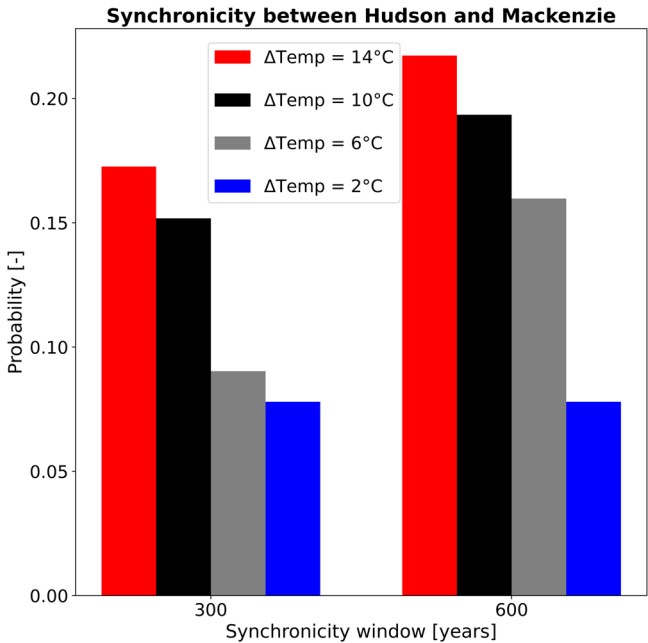

**Fig. 3 | Synchronous Heinrich events from different ice streams of the Laurentide ice sheet.** Probability of synchronous Heinrich events from the Hudson and Mackenzie ice streams. The probability of synchronous changes increases for larger surface temperature perturbations.

mechanism not only leads to phase locking of Heinrich events in the late interstadial phase of Dansgaard-Oeschger cycles, but can also produce synchronous Heinrich events from the Hudson and Mackenzie ice streams. This simulated behaviour is robust under a wide range of forcing scenarios. A distinct advantage of our mechanism is that it is not restricted to marine-terminating ice streams, but applies equally to land-terminating ice streams that only become marine-terminating during the actual Heinrich event. Moreover, our findings indicate that the length of activation and stagnation cycles for present-day ice streams, which are important contributors to sea-level rise, may be controlled by atmospheric perturbations.

## Methods

### Ice sheet-solid Earth model

In our simulations, we employ the coupled ice sheet-solid Earth model mPISM-VILMA, consisting of the modified Parallel Ice Sheet Model (mPISM[30,31]) and the global VIscoelastic Lithosphere and MAntle model (VILMA[32]). The ice dynamics in mPISM are based on the superposition of the shallow ice approximation (SIA) and shallow shelf approximation (SSA), making it suitable to model slow ice flow, fast ice flow as well as grounding-line motion. The model has previously been shown to successfully reproduce the dynamics of internally-driven Heinrich events[26,31]. We use a northern hemispheric setup of mPISM with a horizontal mesh resolution of 10 km. The three-dimensional temperature distribution of the ice sheet is computed based on the enthalpy method[33]. In mPISM, we modified the enthalpy advection to use the c-grid that is also used in PISM for mass advection.

The surge characteristics are strongly affected by the basal sliding parameterisation. In all our simulations, we apply a non-linear Weertmann-type friction law that relates the basal shear stress $\boldsymbol{\tau}_b$ to the basal sliding velocity $\boldsymbol{u}_b$ of the form:

$$\boldsymbol{\tau}_b = -\tau_c \frac{\boldsymbol{u}_b}{u_0^q |\boldsymbol{u}_b|^{1-q}}. \tag{1}$$

Here $q$ is the basal sliding coefficient, which is set to 0.25 in all simulations, $\tau_c$ is the yield stress, and $u_0$ is the threshold velocity at which the magnitude of the basal shear stress equals $\tau_c$. We set $u_0$ to 70 m yr$^{-1}$ for all simulations. The yield stress $\tau_c$ is computed via the Mohr-Coulomb criterion

$$\tau_c = \tan(\phi) N_{till}. \tag{2}$$

Here, $\phi$ represents the till friction angle that is parameterised based on bedrock elevation. The friction angle varies linearly from 15° to 30° for bedrock elevations between -300 m and 400 m and remains constant at the upper or lower limit otherwise. The effective pressure ($N_{till}$) is determined from the ice overburden pressure and the till saturation. Subglacial till water is simulated based on a non-conserving undrained plastic bed model[34]. This means that meltwater is produced and consumed locally, and horizontal transport of meltwater is neglected. To enhance basal sliding for the ice-sheet surges, we spread 50% of the basal heating effect from the sliding in each cell over the four neighbour cells. This results in a faster surge propagation and shorter surge periods that is more in accordance with reconstructions.

The formation of icebergs is approximated using two separate parameterisations. First, we apply so-called Eigencalving[35], where the detachment of icebergs is based on the spreading rates of the ice shelf. In addition, we also apply a thickness calving criterion which removes ice at the calving front that is thinner than 100 m at a maximum rate of one grid cell per time step.

The evolution of the solid earth is modelled by VILMA using a 1D Earth structure. This implicitly assumes that the structure of the solid Earth varies in depth but is otherwise spatially homogeneous. The coupling of mPISM and VILMA is performed every 100 model years and includes the exchange of the ice thickness distribution and glacial isostatic adjustment fields between the two models. As a global model, VILMA requires a global ice thickness distribution as input. Since we focus on the evolution of the Laurentide ice sheet, we keep the ice thickness distribution of all other northern and southern hemispheric ice sheets constant through time.

### Initial state
The initial state for all our mPISM-VILMA simulations was taken at 36 kyr before present from an accelerated Marine Isotope State 3 (MIS3) simulation with the Max Planck Institute for Meteorology Earth System Model (MPI-ESM) coupled to mPISM and VILMA. The state represents MIS3 climate conditions and is characterised by extensive ice sheets such as the Laurentide, Fennoscandian, and Russian ice sheets in the northern hemisphere and the Antarctic ice sheet in the southern hemisphere.

### Dansgaard-Oeschger cycle forcing
All simulations are forced with a time-constant MIS3 background climate. The climate forcing includes ocean temperature and ocean salinity to compute basal melt rates under ice shelves. The atmospheric forcing, consisting of ice surface temperature and surface mass balance fields, is provided through an energy balance model[36]. The surface temperature hereby corresponds to the temperature of the lowest layer in a snow model, restricting ice surface temperature to values below freezing. Applying constant MIS3 forcing ensures that the model drift and corresponding ice volume differences within our simulations remain low (<3%). To superimpose the spatially homogeneous rapid warming and gradual cooling behaviour of Dansgaard-Oeschger cycles as derived from the paleo record, we constructed a forcing function that mimics this behaviour of the following form:

$$\text{forcing}_{do} = \frac{1}{d} \arctan\left( \frac{d \times \sin(2\pi(t_i - t_0)/T)}{1 - d \times \cos(2\pi(t_i - t_0)/T)} \right). \tag{3}$$

Here $d$ represents the distortion factor and is set to 0.75 in all our simulations to reproduce the rapid warming-gradual cooling cycle. $T$ is the length of the Dansgaard-Oeschger cycle. Depending on the simulation, $T$ takes values of 1000, 1500, or 2000 years. The variables $t_i$ and $t_0$ represent the current and the initial time of the model simulation. To ensure the function values of Eq. (3) map between [−1,1], we scale the weights according to:

$$\text{forcing}_{do\_scaled} = \frac{\text{forcing}_{do}}{\max(\text{forcing}_{do})}. \tag{4}$$

It is important to note that the weights of the forcing function sum to zero when integrated over an entire Dansgaard-Oeschger cycle. This means that the background climate is not altered during the simulations and only on time scales shorter than the length of the Dansgaard-Oeschger cycle does the forcing function result in a transient perturbation of the climate forcing. The final forcing functions for the surface temperature and the surface mass balance are described by:

$$A_{smb} = \Delta\text{SMB} \times \text{forcing}_{do\_scaled} \tag{5}$$

$$A_{temp} = \Delta\text{Temp} \times \text{forcing}_{do\_scaled}, \tag{6}$$

where $\Delta$(SMB, Temp) are the prescribed anomaly magnitudes of the respective simulation.

### Coupled simulations
All presented perturbation simulations were integrated for a total of 57,000 years, equivalent to 38 Dansgaard-Oeschger cycles. This resulted in a minimum of 9 Heinrich events for Hudson ice stream and 11 Heinrich events for Mackenzie ice stream. As Mackenzie ice stream is located in a climate close to a threshold under which Heinrich events cannot be maintained anymore[26]. A few simulations exist in which Mackenzie ice stream turns into a persistent ice stream. However, since Mackenzie ice stream predominatly serves the purpose of showing the possibility of synchronous Heinrich events, we deem this less critical. To permit a more robust statistical analysis, the reference simulation was integrated for a total of 123,000 years, corresponding to 82 Dansgaard-Oeschger cycles. To determine whether Heinrich event distributions from the perturbation simulations are significantly different from the Heinrich event distribution of the reference simulation, we used the Kuiper's test. This test is closely related to the more widely-known Kolmogorov–Smirnov test, but has the additional property that it remains invariant under cyclic transformation, making it well-suited to the application of cyclic variations such as our Dansgaard-Oeschger forcing.

### Ensemble simulations with spatially varying forcing
To evaluate the robustness of our results with the idealised forcing, we ran additional experiments with a more realistic and spatially-varying surface temperature and surface mass balance Dansgaard-Oeschger cycle forcing. The forcing is constructed from a MIS3 simulation with a comprehensive Earth System Model that includes interactive ice sheet and solid earth components (see "Methods" section: Initial state). As this model does not exhibit the characteristic Dansgaard-Oeschger cycles of MIS3, we approximate the expected Dansgaard-Oeschger cycle atmospheric forcing as the difference between years with strong Atlantic Meriodional Overturning Circulation (AMOC; >14 Sv) and years with weak AMOC (<12 Sv). Overall, the AMOC varied between -8–18 Sv. Because we still anticipate the forcing magnitudes to be lower than from a full Dansgaard-Oeschger cycle, we perform simulations where we multiply the forcing magnitudes by constants of 1, 2, 3 and 4. The constructed 2D forcing shows similar perturbation magnitudes to the range covered in the spatially homogeneous

simulations (Supplementary Fig. 6). We therefore observe a phase locking behaviour of Hudson ice stream that closely resembles the results of the simulations with the more idealised forcing (Supplementary Fig. 7).

## Data availability
The data required to reproduce all figures in this study have been deposited in the Zenodo database under accession code https://doi.org/10.5281/zenodo.10777268[37].

## Code availability
The PISM base code of version 0.7.3 that was used for all simulations shown in this manuscript is available at Zenodo under https://doi.org/10.5281/zenodo.7541412[38]. The repository details installation instructions and contains information on how to run the model.

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

## Acknowledgements

Clemens Schannwell, Marie-Luise Kapsch and Uwe Mikolajewicz were supported by the German Federal Ministry of Education and Research

(BMBF) as a Research for Sustainability initiative (FONA) through the PalMod project under grant numbers 01LP1915C, 01LP1916A and 01LP1917B. This work used resources of the Deutsches Klimar-echenzentrum (DKRZ) granted by its Scientific Steering Committee (WLA) under project ID ba0989. Development of PISM is supported by NSF grants PLR-1644277 and PLR-1914668 and NASA grants NNX17AG65G and 20-CRYO2020-0052. We thank Jürgen Bader for comments on an earlier version of the manuscript.

## Author contributions

C.S. and U.M. conceived the study. F.Z. developed the initial model setup with input from M.L.K.; C.S. performed the experiments and analysed the data. The paper was written by C.S. with input from M.L.K., U.M. and F.Z.

## Funding

## Competing interests

The authors declare no competing interests.
