## [Peer Review File · Nature Communications]

A mechanism for reconciling the synchronisation of Heinrich events and Dansgaard-Oeschger cyclesReviewer #1 (Remarks to the Author):

This study suggests a new hypothesis for Heinrich Events—mass iceberg evacuations from the Laurentide ice sheet. The cause of Heinrich Events have long been debated because they occur during the coldest portions of the glacial cycle. The original hypothesis for Heinrich Events, proposed by Doug MacAyeal involves a two-stage process called the binge-purge cycle where the ice is first frozen to its bed, but then reaches the pressure melting point at the base. The transition to the melting point at the base results in a surging behavior of the ice sheet. Although this hypothesis resulted in a time scale for Heinrich Events roughly comparable to that observed for Heinrich Events, the predicted Heinrich Events were not synchronized with the DO cycles. Subsequent hypotheses have largely revolved around the fact that paleo-proxies indicate that ocean warming associated with DO events precedes Heinrich Events and these subsequent studies have assumed that the warming ocean can act as a trigger for Heinrich Events.

The authors show here that the original binge-purge hypothesis can be resuscitated if temperature and snowfall are taken to be a function of DO events. In the authors model, this results in a Heinrich event cycle that is phase locked to the DO events. The phase locking proposed here seems similar to the phase locking found by Mann et al., (2021), although that study assumed an ocean trigger. Moreover, the novelty of this is the syncing of the binge-purge cycle with the DO events. But the classic binge-purge hypothesis proposed seems like it is still at the heart of the mechanism proposed.

Overall, I think the manuscript is generally well written and put together. I include a couple of suggestions and questions below.

1. Some questions about physics and mechanisms. What causes iceberg calving in the current model? To be more precise, during the purge phase of the DO cycles the ice sheet bed reaches the pressure melting point. This causes the ice stream to surge. But this could cause the ice stream to thin and advance, perhaps forming an ice shelf? Presumably the same mechanism would work on land-terminating ice streams, but would not produce an armada of icebergs? Fundamentally, what causes the ice to break and the deluge of calving events? The surging part of the model is clear to me, but surging glaciers observed today don't typically create armadas of icebergs even when they terminate in the ocean. Figure 2 looks like a small ice shelf is forming, but what controls the extent of the ice shelf?
2. A related question is how do the authors treat sliding, till deformation and the thermodynamics of the bed? Sliding, heat flow and the dynamics at the ice-till interface seems crucial to the model and it would be helpful to see this described in the methods section with a little bit more detail.
3. I don't think it is entirely true that other mechanisms proposed aren't able to reproduce behavior of the non-Hudson ice streams. The ocean forcing hypothesis suggests widespread ocean warming which would trigger widespread retreat. The retreat of Greenland and other glaciers driven by ocean temperatures is analogous to what is observed currently. Where it seems the ocean forced model will result in significant differences is that the model proposed by the authors will apply to both terrestrial and marine margins. This is hinted at in the paper, but not elaborated on. I would encourage the authors to take readers through the evidence that land-terminating ice streams also surged at the exact same time as marine-terminating ice streams. I think this would be powerful evidence that it isn't an ocean signal that is driving these events and would provide strong support for the authors modified binge-purge model.
4. My next comment relates to how the authors have driven their model. The DO events observed in the record are not periodic. There have been some hypothesis that DO events are stochastic poisson processes. What I would like to know is whether the phase locking is a consequence of the assumption

that the DO events are periodic and occur quasi-regularly. What happens if the onset of DO events are set according to the observed records (or stochastically) with long gaps between some of the DO events. Do you still avoid Heinrich Events during the long gaps? Does the phase locking only occur in a small portion of the record? Or does it occur across large portions of the record?

5. This is a minor point, but why not use the temperature and mass balance perturbation associated with DO events inferred from the Greenland ice sheet ice cores? This seems like it would be especially useful in a simulation driven with the timing and magnitude of observed DO events. I realize that this would not necessarily apply to the Laurentide, but it would provide a magnitude.

Mann, L. E., Robel, A. A., & Meyer, C. R. (2021). Synchronization of Heinrich and Dansgaard-Oeschger events through ice-ocean interactions. *Paleoceanography and Paleoclimatology*

Reviewer #2 (Remarks to the Author):

This manuscript on "A mechanism for reconciling the synchronization of Heinrich events and Dansgaard Oeschger cycles" discussed a new modeling-based theory for millennial-scale glacial climate variability and the interactions between two different modes of variability. The central argument is that atmospheric processes at the surface of the Laurentide ice sheet allow for synchronization between internally generated ice sheet variability (Heinrich events) and ocean variability (D-O events). This synchronization potentially helps resolve several outstanding questions in the literature including the occurrence of Heinrich events during D-O stadials and the activation of many Laurentide ice streams during some Heinrich events.

The particular idea here (synchronization of Heinrich and DO events through surface temperature and SMB) is novel, though it has unacknowledged similarities to previous work on Heinrich-DO synchronization. The manuscript does a good job explaining why these mechanisms explain certain enigmas within the paleoclimate records. It does so by building on well established models and ideas from climate and ice sheet dynamics.

Major concerns:

1. This manuscript argues that certain aspects of the results improve on prior theories in ways which are not clearly supported by the model results.

(a) In the abstract and later, a distinction is made between a "trigger mechanism" and "preconditioning". It is not clear to me how these are different. It could be argued that the atmospheric warming "triggers" the Heinrich event by causing the basal thermal state to warm. To me, this difference seems to be more rhetorical than substantive. A related point is that in systems of synchronized oscillators causality isn't straightforward. The oscillations in the two systems (ice sheet and ocean in this case) would happen regardless of the interaction between them, but both systems interact with each other to produce a situation in which the timing of the events is consistent. One does not "cause" the other, but their timing is related.

(b) In the abstract and elsewhere, it is argued that the mechanism put forward in this study "provides a simple explanation for the observational evidence of synchronous Heinrich events from different ice sheets". The MacKenzie and Hudson Strait ice streams are in the same ice sheet, so its no clear that this is in fact demonstrated in this study. Given that the model configuration in question seemingly includes the Fennoscandian ice sheet, it would seem that a similar analysis could be done to determine whether the atmospheric mechanism also produces synchronization with Fennoscandian ice streams.

However, if such evidence is lacking, then it should not be claimed. Perhaps more accurate would be to say that it "provides a simple explanation for the observational evidence of synchronous discharge events from different ice streams within the Laurentide Ice Sheet".

2. The central idea in this study is the most similar to a recent study by Mann et al. (2021 in *Paleoclimatology and Paleoceanography*) which is also able to explain the synchronization of Heinrich and DO events, but instead using ice-ocean interactions to generate synchronization between a flow line glacier model and a simple ocean model. Clearly this manuscript proposes a different synchronization method and uses more realistic models, but that prior study provides many potential points of comparison (including more comprehensive parameter sweeps). One thing that is not clear is whether the Mann et al. ocean-based mechanism for synchronization, if included in a more complex model like this one, could also explain all the aspects of the paleoclimate record that the authors focus on in this study. The paper would benefit from a more extensive citation and comparison to the previous study by Mann et al.

3. One surprising result is that surface temperature perturbations from DO events have any significant impact on ice sheet sliding on time scales of a few thousand years. The diffusion time scale of heat through thick ice is tens of thousands of years. This is why in the original papers on the B-P theory for Heinrich events, MacAyeal discounts the role of surface temperature changes in driving Heinrich events because they would take too long to diffuse through the ice sheet. It would seem that a more robust explanation is needed here, particularly with some plots showing the influence of adding the atmospheric perturbation on the basal heat budget of ice streams.

Minor comments:

Line 87: In what sense do you mean "statistically significant different"? What metrics characterizing the Heinrich event are you measuring and how is that difference measured?

Line 90: Why does it seem that the SMB perturbation has more of an effect than the temperature perturbation?

Line 93: Increased surface temperature will increase snow accumulation at high elevation but also increase surface melting at low elevations? Is the latter not important at the edges of the ice sheet?

Line 95: What is the mechanism explaining why "The onset of the Heinrich event is, however, delayed into the early phase of the transition into the stadial of the Dansgaard-Oeschger cycle"

Line 99: Similarly, can you explain why it takes 3-5 D-O cycles before the next Heinrich events occurs? Is it simply that H events are paced internally by ice streams? Do the Hudson Strait and MacKenzie ice streams have the same natural periodicity for discharge events?

Line 140: Can you say with significance that an increase from 8% to 17% or to 22% would not just happen by chance? How many more Heinrich events is this? Is it more than one?

Line 158: What makes this an advantage? Why do you need this mechanism to apply to land-terminating glaciers? It wouldn't help explain anything about the marine sediment record which is only informed by marine-terminating glacier discharge.

L163: related to point #3 above, it would be helpful to explain here how internal temperature and SMB are modeled in mPISM-VILMA since those processes are central to the mechanism in this study.

Reviewer #3 (Remarks to the Author):

Review of the manuscript "A mechanism for reconciling the synchronisation of Heinrich events and Dansgaard-Oeschger cycles" by Dr Schannwell and colleagues.

This paper presents a novel mechanism that is introduced to explain the synchronous occurrence of Heinrich events and the stadial phase of Dansgaard-Oeschger cycles during the last glacial period. Heinrich events, quasi-episodic iceberg discharge events from the Laurentide ice sheet into the North Atlantic, have been a significant feature of the northern hemispheric climate during the last ice age cycle and is a difficult process to study because of the large range of temporal and spatial scales that are involved and the small number of Heinrich events observed. Using simulations with a coupled ice sheet-solid earth model, the researchers present a novel mechanism that eliminates the need for a trigger mechanism during the stadial phase. Instead, it is proposed that the interstadial phase of the Dansgaard-Oeschger cycle, characterised by atmospheric warming, preconditions the ice sheet for a Heinrich event, causing it to occur during the subsequent stadial phase.

This paper re-visits the binge-purge hypothesis and presents a fresh re-interpretation on the mechanisms behind Heinrich events. While I personally resonate with the subsurface ocean warming marine ice sheet instability hypothesis which occurs during stadials, it is important to provide other possible theories that may help explain a dominant feature of ice age climate. I feel the paper is well written and the study rests on some solid previous work by the group. I would like to see some of my concerns addressed which I list below.

General Structure of the paper:

I like Figure 2 and I think this nice graphic should be the highlight of the paper. I would suggest moving this to Figure 1. It should not be too hard to rework the text on pages 4 and 5. Start with Pg 5 L92. I have more specific comments on the figure below.

Specific comments:

Figure 1:

Why is there a colour wheel in the "Unforced reference simulation" in figure 1a . Is this not forced with constant MIS3 forcing and so should be blank? Or have I missed something?

The current Figure 1 could be enhanced with another schematic diagram beside the top panel "Unforced reference simulation" (to the left , illustrating how the ice volume builds as a curve spirals and then collapses with a Heinrich event. Also it was not initially clear what all the thin black lines were indicating (the ensemble of runs) and how many revolutions they made. Do all the thin lines start at 0 time at some small ice volume? It will take the reader some time to see what is happening so a simpler schematic of one simulation with text (possibly) highlighting HEs and DO stadial and Interstadial periods.

The matrix of plots in figure 1 has Δ SMB vs Δ Temp, shouldn't this be A_{smb} vs A_{temp} (the amplitudes given from the equations (3 and 4) in the Methods section. This could be changed and pointed out in the figure caption.

What does the white significance level indicate (< 90th percentile) , maybe state this.

Figure 2:

It would be useful to highlight that the insets to figure 2 are from equation 1 in the Methods section

and the colours on the curves represent the derivative of the temperature and mass balance changes.

To most people I think they would associate the point in Figure 2c as early interstadial. It is so close to the D-O warming event that I don't think one can possibly argue that this is a stadial condition.

Pg. 3 L 51-54: Is it possible to infer evidence of ice shelves from the DeVernal et al 2000 paper on LGM. Is there a strong statement in this paper about no ice shelves existing along the Hudson Strait Ice Stream before H1?

Pg 4: L67-68: On explaining the observed characteristics of the paleo record: The atmospheric forcing function (Equation 1) does not have the same quick drop off at the end of the interstadial as seen in the ice core records (e.g. Rasmussen et al 2014). In particular if you look at GI-13 the interstadial period remains elevated for some time (800 years) and then quickly drops off. The results here possibly work here with H5, but if you look at GI-9 the interstadial is short (< 500 years). The interstadials previous to the other Heinrich events do not fit well into the context of this paper and may have other origins other than the Hudson's Bay Ice Stream. It would be difficult to say anything using observations to corroborate the findings of this study.

Pg 4. L80: This would be a good place to highlight the amplitude from the methods (A_{sbm} and A_{temp}) and refer to the methods.

Pg5 L88-89:

If you assume that there are Heinrich events at 30, 39 and 47.5 kaBP you would argue that there is a series of decreasing interstadial lengths previous to the H-Events as observed from the ice core data (a sort of damped D-O oscillation).

I would like to see what kind of effect would result from a damped harmonic oscillator (e.g. if you added an $\exp(-t/\tau)$ in front of the arctan term in Methods equation 1. From the results presented, I assume that the synchronisation would decrease as the amplitude goes down. So I have a hard time seeing how this could be corroborated with observations. I didn't see any strong attempt to argue how the model results compare with observations of D-O variability.

Zieman et al 2019 (CP) include oceanic forcing. Without the influence of ocean effects it is really hard to argue that this is "the" mechanism that causes H-Events.

Pg5 L96-97:

Is this consistent with Zieman et al 2019? How much is this in mSv (50)?

Pg5 102-104: These results suggest that the short D-O cycles would have an H-Event close to the end of the interstadial, is this consistent with observations.

Pg 6: L 122-123: Figure 1 suggests that the SMB is the primary driver from figure 1 if I read the significance correctly.

Pg 14: L219-224: What is the range of the AMOC in the MIS3 simulation?

In the extended data figures, what is the change in radial ice volume in meters of RSL (it is only given in m^3).

Response to the reviewers

We thank all referees for their thoughtful and thorough reviews of our paper. We appreciate you taking the time to complete these reviews and welcome your helpful comments. In the following we address their concerns point by point. Throughout this response to review document referee comments are provided in regular, non- italic font text, our response comments are provided in red font.

To accomodate the main comments of the reviewers, we have made the following main changes to the manuscript:

- We have substantially expanded the model description section to provide a more complete picture of the implementation of key processes.
- We have added four more simulations in which we vary the DO cycle length stochastically from one cycle to the other.
- Reworked the general structure of the manuscript by swapping the order of Figure 1 and Figure 2 as well as text adjustments reflecting the modified Figure order.

Reviewer 1

Reviewer Point P 1.1 — This study suggests a new hypothesis for Heinrich Events—mass iceberg evacuations from the Laurentide ice sheet. The cause of Heinrich Events have long been debated because they occur during the coldest portions of the glacial cycle. The original hypothesis for Heinrich Events, proposed by Doug MacAyeal involves a two-stage process called the binge-purge cycle where the ice is first frozen to its bed, but then reaches the pressure melting point at the base. The transition to the melting point at the base results in a surging behavior of the ice sheet. Although this hypothesis resulted in a time scale for Heinrich Events roughly comparable to that observed for Heinrich Events, the predicted Heinrich Events were not synchronized with the DO cycles. Subsequent hypotheses have largely revolved around the fact that paleo-proxies indicate that ocean warming associated with DO events precedes Heinrich Events and these subsequent studies have assumed that the warming ocean can act as a trigger for Heinrich Events.

The authors show here that the original binge-purge hypothesis can be resuscitated if temperature and snowfall are taken to be a function of DO events. In the authors model, this results in a Heinrich event cycle that is phase locked to the DO events. The phase locking proposed here seems similar to the phase locking found by Mann et al., (2021), although that study assumed an ocean trigger. Moreover, the novelty of this is the syncing of the binge-purge cycle with the DO events. But the classic binge-purge hypothesis proposed seems like it is still at the heart of the mechanism proposed.

Overall, I think the manuscript is generally well written and put together. I include a couple of suggestions and questions below.

Reply: We thank the reviewer for the positive assessment of our manuscript.

Reviewer Point P 1.2 — Some questions about physics and mechanisms. What causes iceberg calving in the current model? To be more precise, during the purge phase of the DO cycles the ice sheet bed reaches the pressure melting point. This causes the ice stream to surge. But this could cause the ice stream to thin and advance, perhaps forming an ice shelf? Presumably the same mechanism would work on land-terminating ice streams, but would not produce an armada of icebergs? Fundamentally, what causes the ice to break and the deluge of calving events? The surging part of the model is clear to me, but surging glaciers observed today don't typically create armadas of icebergs even when they terminate in the ocean. Figure 2 looks like a small ice shelf is forming, but what controls the extent of the ice shelf?

Reply: We have added a description of how iceberg calving is parameterised in our simulations. The reviewer is correct that this mechanism works also for land-terminating ice streams, but they would not produce icebergs but a liquid freshwater flux that would be transported to the ocean via rivers. During the build-up phase, Mackenzie is land-terminating. For the Hudson ice stream, a small ice shelf typically forms during a Heinrich event, but its existence is short-lived and the ice-stream front retreats to its pre-surge position. This is also shown in this video from our previous paper: Video link. In comparison, Mackenzie ice stream is land-terminating during the build-up phase and only becomes marine terminating with a small floating ice tongue during the actual Heinrich event.

Reviewer Point P 1.3 — A related question is how do the authors treat sliding, till deformation and the thermodynamics of the bed? Sliding, heat flow and the dynamics at the ice-till interface seems crucial to the model and it would be helpful to see this described in the methods section with a little bit more detail.

Reply: We agree that this strongly influences the surge characteristics. We have substantially expanded the Methods section to describe this in more detail.

Reviewer Point P 1.4 — I don't think it is entirely true that other mechanisms proposed aren't able to reproduce behavior of the non-Hudson ice streams. The ocean forcing hypothesis suggests widespread ocean warming which would trigger widespread retreat. The retreat of Greenland and other glaciers driven by ocean temperatures is analogous to what is observed currently. Where it seems the ocean forced model will result in significant differences is that the model proposed by the authors will apply to both terrestrial and marine margins. This is hinted at in the paper, but not elaborated on. I would encourage the authors to take readers through the evidence that land-terminating ice streams also surged at the exact same time as marine-terminating ice streams. I think this would be powerful evidence that it isn't an ocean signal that is driving these events and would provide strong support for the authors modified binge-purge model.

Reply: We have expanded the paragraph why we think it is an advantage that our mechanism works equally well for permanently marine-terminating ice stream, such as Hudson ice stream, as well as for ice streams that are land-terminating and only become marine-terminating during the actual Heinrich event. We have added evidence of surging from land-terminating ice streams of the Laurentide ice sheet as well as evidence from synchronous surges of mountain glaciers, a modern-day analogue for Heinrich events.

Reviewer Point P 1.5 — My next comment relates to how the authors have driven their model. The DO events observed in the record are not periodic. There have been some hypothesis that DO

events are stochastic poisson processes. What I would like to know is whether the phase locking is a consequence of the assumption that the DO events are periodic and occur quasi-regularly. What happens if the onset of DO events are set according to the observed records (or stochastically) with long gaps between some of the DO events. Do you still avoid Heinrich Events during the long gaps? Does the phase locking only occur in a small portion of the record? Or does it occur across large portions of the record?

Reply: Our analysis here focuses primarily on the MIS3 period during which five of the six recorded Heinrich events occurred. During this period, DO cycles are fairly regular and prolonged gaps between DO cycles are not present. If gaps would extend to longer than 5,000 - 7,000 years, our model would produce a Heinrich event even in the absence of a DO cycle. To further investigate the robustness of our results to stochastic variability in the DO cycle, we performed four additional simulations in which the DO cycle length is randomly varied between 1,000 and 2,000 years (in 100 year intervals). These simulations show an expected weakened phase locking signal in response to the increased noise level (see Extended Figure 8 in the revised manuscript). Still, the vast majority of the events still occurs during the cooling phase of the DO cycle. Interestingly, the onset of many Heinrich events in these simulations happens much deeper into the stadial phase which is more in accordance with observations. It is important to note that not all of the recorded Heinrich events occurred during the stadial phase of DO cycles.

Reviewer Point P 1.6 — This is a minor point, but why not use the temperature and mass balance perturbation associated with DO events inferred from the Greenland ice sheet ice cores? This seems like it would be especially useful in a simulation driven with the timing and magnitude of observed DO events. I realize that this would not necessarily apply to the Laurentide, but it would provide a magnitude.

Reply: The chosen magnitude ranges are based on Greenland ice core reconstructions as mentioned in Line 98 in our initial submission. Because there are substantial variations in signal magnitude from DO cycle to DO cycle, we chose to rather encompass this range with our ensemble approach rather than choose a specific value. As pointed out by the reviewer and shown in our 2D model forcing (Extended Figure 6) in the initial submission, the magnitudes over the Laurentide ice sheet are much lower than what has been recorded in the Greenland ice cores.

Reviewer 2

Reviewer Point P 2.1 — This manuscript on “A mechanism for reconciling the synchronization of Heinrich events and Dansgaard Oeschger cycles” discussed a new modeling-based theory for millennial-scale glacial climate variability and the interactions between two different modes of variability. The central argument is that atmospheric processes at the surface of the Laurentide ice sheet allow for synchronization between internally generated ice sheet variability (Heinrich events) and ocean variability (D-O events). This synchronization potentially helps resolve several outstanding questions in the literature including the occurrence of Heinrich events during D-O stadials and the activation of many Laurentide ice streams during some Heinrich events.

The particular idea here (synchronization of Heinrich and DO events through surface temperature

and SMB) is novel, though it has unacknowledged similarities to previous work on Heinrich-DO synchronization. The manuscript does a good job explaining why these mechanisms explain certain enigmas within the paleoclimate records. It does so by building on well established models and ideas from climate and ice sheet dynamics.

Reply: We thank the reviewer for the positive assessment of our manuscript.

Reviewer Point P 2.2 — This manuscript argues that certain aspects of the results improve on prior theories in ways which are not clearly supported by the model results.

(a) In the abstract and later, a distinction is made between a "trigger mechanism" and "preconditioning". It is not clear to me how these are different. It could be argued that the atmospheric warming "triggers" the Heinrich event by causing the basal thermal state to warm. To me, this difference seems to be more rhetorical than substantive. A related point is that in systems of synchronized oscillators causality isn't straightforward. The oscillations in the two systems (ice sheet and ocean in this case) would happen regardless of the interaction between them, but both systems interact with each other to produce a situation in which the timing of the events is consistent. One does not "cause" the other, but their timing is related.

Reply: We have rephrased this in the revised version of the manuscript. We now highlight that the underlying binge-purge mechanism preconditions the ice stream for a Heinrich event. The final trigger that determines the onset of the rapid sliding is however provided by the DO cycle forcing. As a result, this means that Heinrich events occur in our model even without the DO cycle forcing, but the timing and periodicity of the events differs between the unforced and forced Heinrich event simulations.

Reviewer Point P 2.3 — (b) In the abstract and elsewhere, it is argued that the mechanism put forward in this study "provides a simple explanation for the observational evidence of synchronous Heinrich events from different ice sheets". The MacKenzie and Hudson Strait ice streams are in the same ice sheet, so its no clear that this is in fact demonstrated in this study. Given that the model configuration in question seemingly includes the Fennoscandian ice sheet, it would seem that a similar analysis could be done to determine whether the atmospheric mechanism also produces synchronization with Fennoscandian ice streams. However, if such evidence is lacking, then it should not be claimed. Perhaps more accurate would be to say that it "provides a simple explanation for the observational evidence of synchronous discharge events from different ice streams within the Laurentide Ice Sheet".

Reply: Since the Fennoscandian ice sheet is held fixed in our simulations, we follow the recommendation from the reviewer and removed references to other ice sheets from the manuscript.

Reviewer Point P 2.4 — The central idea in this study is the most similar to a recent study by Mann et al. (2021 in Paleoclimatology and Paleoceanography) which is also able to explain the synchronization of Heinrich and DO events, but instead using ice-ocean interactions to generate synchronization between a flow line glacier model and a simple ocean model. Clearly this manuscript proposes a different synchronization method and uses more realistic models, but that prior study provides many potential points of comparison (including more comprehensive parameter sweeps). One thing that is not clear is whether the Mann et al. ocean-based mechanism for synchronization, if

included in a more complex model like this one, could also explain all the aspects of the paleoclimate record that the authors focus on in this study. The paper would benefit from a more extensive citation and comparison to the previous study by Mann et al.

Reply: We have added a citation to Mann et al as well as a paragraph to the introduction that acknowledges that phase locking has also been achieved in simple coupled models of two autonomous oscillators through a weak coupling if the coupling is strong enough to overcome the natural frequencies of the two oscillators.

Reviewer Point P 2.5 — One surprising result is that surface temperature perturbations from DO events have any significant impact on ice sheet sliding on time scales of a few thousand years. The diffusion time scale of heat through thick ice is tens of thousands of years. This is why in the original papers on the B-P theory for Heinrich events, MacAyeal discounts the role of surface temperature changes in driving Heinrich events because they would take too long to diffuse through the ice sheet. It would seem that a more robust explanation is needed here, particularly with some plots showing the influence of adding the atmospheric perturbation on the basal heat budget of ice streams.

Reply: The reviewer is correct that the diffusion timescale is much longer than the few hundred years of warming during a DO cycle. Rather, the driving process here is that the surface warming results in an enhanced ice deformation because the ice deformation law is dependent on ice temperature. This in turn raises the internal frictional heat generation and slightly warms the ice temperature at the ice base. The effect is demonstrated in belows Hovmoeller plots.

Minor

Reviewer Point P 2.6 — Line 87: In what sense do you mean "statistically significant different"? What metrics characterizing the Heinrich event are you measuring and how is that difference measured?

Reply: We have added a short explanation what is being compared and what statistical test we use. We also added a reference to the Methods section where this is also detailed.

Reviewer Point P 2.7 — Line 90: Why does it seem that the SMB perturbation has more of an effect than the temperature perturbation?

Reply: We showed in an earlier publication [Schannwell et al., 2023] that Heinrich event periodicity is primarily affected by surface mass balance because of its climatic setting which is characterised by cold temperatures (well below -20°C) and by surface mass balance values between 0.1 - 0.15 m/yr. In such a regime, the system reacts to surface mass balance perturbations much more because the relative magnitude of the perturbation is much larger than for the surface temperature. We have added a citation to our previous paper.

Reviewer Point P 2.8 — Line 93: Increased surface temperature will increase snow accumulation at high elevation but also increase surface melting at low elevations? Is the latter not important at the edges of the ice sheet?

Figure 1: (a) Shows flowline along Hudson ice stream that is shown in (b-f) as Hovmoeller plots for the first 600 simulation years. Δ refers to difference between the maximum temperature, minimum SMB perturbation simulation and the unforced reference simulation.

Reply: In reality this is correct, but our applied perturbations here are spatially homogeneous and act independently. We have added missing information about the SMB computation to the Methods section (see point below).

Reviewer Point P 2.9 — Line 95: What is the mechanism explaining why “The onset of the Heinrich event is, however, delayed into the early phase of the transition into the stadial of the Dansgaard-Oeschger cycle”

Reply: We have rephrased this sentence to highlight that it is not sufficient for a single location to reach pressure-melting-point and permit rapid basal sliding, but the warm-based region first has to reach a certain size before a Heinrich event is triggered. The lag was referring to the time difference between reaching pressure-melting-point and for the region to grow large enough to trigger a Heinrich event.

Reviewer Point P 2.10 — Line 99: Similarly, can you explain why it takes 3-5 D-O cycles before the next Heinrich events occurs? Is it simply that H events are paced internally by ice streams? Do the Hudson Strait and MacKenzie ice streams have the same natural periodicity for discharge events?

Reply: This is because the natural frequency dictated through the binge-purge is roughly of that order. Of course the natural frequency is modified by different background climate conditions and other perturbations. As shown in Schannwell et al. [2023], Hudson (7,200 years) and Mackenzie ice stream

(4,200 years) indeed do have very different natural surge frequencies. We have added a short mention of this to the main text.

Reviewer Point P 2.11 — Line 140: Can you say with significance that an increase from 8% to 17% or to 22% would not just happen by chance? How many more Heinrich events is this? Is it more than one?

Reply: We have replaced “significant” with “substantial” to remove the statistical connotation of the sentence. We also added the information of how many more synchronous Heinrich events occurred during the simulations. Provided that the signal is monotonously increasing with higher temperature perturbations and consistent across search windows from 100 years to 800 years, we are very confident that the findings are robust.

Reviewer Point P 2.12 — Line 158: What makes this an advantage? Why do you need this mechanism to apply to land-terminating glaciers? It wouldn't help explain anything about the marine sediment record which is only informed by marine-terminating glacier discharge.

Reply: We have clarified that we think that our mechanism provides an advantage in that it works equally well for permanently marine-terminating ice streams like the Hudson ice stream, but also for ice streams that are land-terminating and only become marine-terminating during the actual Heinrich event.

Reviewer Point P 2.13 — L163: related to point #3 above, it would be helpful to explain here how internal temperature and SMB are modeled in mPISM-VILMA since those processes are central to the mechanism in this study.

Reply: We have added the missing information to the Methods section.

Reviewer 3

Reviewer Point P 3.1 — This paper presents a novel mechanism that is introduced to explain the synchronous occurrence of Heinrich events and the stadial phase of Dansgaard-Oeschger cycles during the last glacial period. Heinrich events, quasi-episodic iceberg discharge events from the Laurentide ice sheet into the North Atlantic, have been a significant feature of the northern hemispheric climate during the last ice age cycle and is a difficult process to study because of the large range of temporal and spatial scales that are involved and the small number of Heinrich events observed. Using simulations with a coupled ice sheet-solid earth model, the researchers present a novel mechanism that eliminates the need for a trigger mechanism during the stadial phase. Instead, it is proposed that the interstadial phase of the Dansgaard-Oeschger cycle, characterised by atmospheric warming, preconditions the ice sheet for a Heinrich event, causing it to occur during the subsequent stadial phase.

This paper re-visits the binge-purge hypothesis and presents a fresh re-interpretation on the mechanisms behind Heinrich events. While I personally resonate with the subsurface ocean warming

marine ice sheet instability hypothesis which occurs during stadials, it is important to provide other possible theories that may help explain a dominant feature of ice age climate. I feel the paper is well written and the study rests on some solid previous work by the group. I would like to see some of my concerns addressed which I list below.

Reply: We thank the reviewer for the positive assessment of our manuscript.

Reviewer Point P 3.2 — General Structure of the paper:

I like Figure 2 and I think this nice graphic should be the highlight of the paper. I would suggest moving this to Figure 1. It should not be too hard to rework the text on pages 4 and 5. Start with Pg 5 L92. I have more specific comments on the figure below.

Reply: Yes, we agree with that and have swapped the order of Fig. 1 and Fig. 2 as well as adjusted the text to reflect the adjusted Figure order.

Minor comments

Reviewer Point P 3.3 — Figure 1: Why is there a colour wheel in the “Unforced reference simulation” in figure 1a. Is this not forced with constant MIS3 forcing and so should be blank? Or have I missed something?

Reply: The reviewer is correct. We removed the colour wheel.

Reviewer Point P 3.4 — The current Figure 1 could be enhanced with another schematic diagram beside the top panel “Unforced reference simulation” (to the left, illustrating how the ice volume builds as a curve spirals and then collapses with a Heinrich event. Also it was not initially clear what all the thin black lines were indicating (the ensemble of runs) and how many revolutions they made. Do all the thin lines start at 0 time at some small ice volume? It will take the reader some time to see what is happening so a simpler schematic of one simulation with text (possibly) highlighting HEs and DO stadial and Interstadial periods.

Reply: We have added an annotated explainer subplot to the revised Figure.

Reviewer Point P 3.5 — The matrix of plots in figure 1 has ΔSMB vs ΔTemp , shouldn't this be A_{smb} vs A_{temp} (the amplitudes given from the equations (3 and 4) in the Methods section. This could be changed and pointed out in the figure caption.

Reply: Since we think that ΔSMB vs ΔTemp are better understandable for the reader, we decided to swap symbols in equations 3 and 4 instead.

Reviewer Point P 3.6 — What does the white significance level indicate ($< 90\text{th}$ percentile) , maybe state this.

Reply: We have added this to the Figure caption.

Reviewer Point P 3.7 — Figure 2: It would be useful to highlight that the insets to figure 2 are from equation 1 in the Methods section and the colours on the curves represent the derivative of the temperature and mass balance changes.

Reply: We have added this information to the revised Figure caption.

Reviewer Point P 3.8 — To most people I think they would associate the point in Figure 2c as early interstadial. It is so close to the D-O warming event that I don't think one can possibly argue that this is a stadial condition.

Reply: We have changed the term to "late interstadial" in the revised version.

Reviewer Point P 3.9 — Line 51-54: Is it possible to infer evidence of ice shelves from the DeVernal et al 2000 paper on LGM. Is there a strong statement in this paper about no ice shelves existing along the Hudson Strait Ice Stream before H1?

Reply: We rephrased that sentence to state that "there is no firm evidence of the existence of such a persistent ice shelf"

Reviewer Point P 3.10 — L67-68: On explaining the observed characteristics of the paleo record: The atmospheric forcing function (Equation 1 Methods does not have the same quick drop off at the end of the interstadial as seen in the ice core records (e.g. Rasmussen et al 2014). In particular if you look at GI-13 the interstadial period remains elevated for some time (800 years) and then quickly drops off. The results here possibly work here with H5, but if you look at GI-9 the interstadial is short (< 500 years). The interstadials previous to the other Heinrich events do not fit well into the context of this paper and may have other origins other than the Hudson's Bay Ice Stream. It would be difficult to say anything using observations to corroborate the findings of this study.

Reply: We are aware that our forcing function is idealised. To capture the effect of temporally varying DO cycles, we performed additional simulations in which the DO cycle length is randomly varied between 1,000 and 2,000 years (in 100 year intervals).

Reviewer Point P 3.11 — L80: This would be a good place to highlight the amplitude from the methods (A_{smb} and A_{temp}) and refer to the methods.

Reply: Yes, we added this.

Reviewer Point P 3.12 — If you assume that there are Heinrich events at 30 , 39 and 47.5 kaBP you would argue that there is a series of decreasing interstadial lengths previous to the H-Events as observed from the ice core data (a sort of damped D-O oscillation). I would like to see what kind of effect would result from a damped harmonic oscillator (e.g if you added an $\exp(-t/\tau)$ in front of the arctan term in Methods equation 1. From the results presented , I assume that the synchronisation would decrease as the amplitude goes down. So I have a hard time seeing how this could be corroborated with observations. I didn't see any strong attempt to argue how the model results compare with observations of D-O variability.

Reply: We agree that the synchronisation will decrease with a decreasing perturbation magnitude. As mentioned above, we did add simulations where we investigated the effect of temporally varying DO cycle lengths. However, from the modelling perspective there is no strong evidence that DO cycles are

the result of a damped oscillator (see Klockmann et al. [2018], Vettoretti and Peltier [2018], Vettoretti et al. [2022]).

Reviewer Point P 3.13 — Ziemeń et al 2019 (CP) include oceanic forcing. Without the influence of ocean effects it is really hard to argue that this is “the” mechanism that causes H-Events.

Reply: We have added to the Methods section that we also force the model with constant ocean conditions. Please note that we tested the sensitivity to the ocean forcing in our model setup in Schannwell et al. [2023] and found that Heinrich events from the Hudson ice stream were unaffected by changes in ocean temperature or sea level forcing.

Reviewer Point P 3.14 — L96-97: Is this consistent with Ziemeń et al 2019? How much is this in mSv (50)?

Reply: Yes this is consistent with Ziemeń et al. 2019. We added the peak discharge rate to the sentence.

Reviewer Point P 3.15 — L102-104: These results suggest that the short D-O cycles would have an H-Event close to the end of the interstadial, is this consistent with observations.

Reply: For some of the recorded Heinrich events such as HE6 and HE4, it is consistent, but for others it is not. The aim of our study is however more a proof of concept of the mechanism rather than aiming to reproduce the observational record exactly. Having said this, we believe that with more model tuning, it is certainly possible to match the observational record better than we do with our idealised forcings.

Reviewer Point P 3.16 — L122-123: Figure 1 suggests that the SMB is the primary driver from figure 1 if I read the significance correctly.

Reply: Yes, this is correct. See reply to Reviewer 2.

Reviewer Point P 3.17 — L219-224: What is the range of the AMOC in the MIS3 simulation? In the extended data figures, what is the change in radial ice volume in meters of RSL (it is only given in m³).

Reply: We have added these numbers to the revised manuscript.

References

- M. Klockmann, U. Mikolajewicz, and J. Marotzke. Two amoc states in response to decreasing greenhouse gas concentrations in the coupled climate model mpi-esm. *Journal of Climate*, 31(19):7969–7984, Oct. 2018. ISSN 1520-0442. doi: 10.1175/jcli-d-17-0859.1. URL <http://dx.doi.org/10.1175/JCLI-D-17-0859.1>.
- C. Schannwell, U. Mikolajewicz, F. Ziemeń, and M.-L. Kapsch. Sensitivity of heinrich-type ice-sheet surge characteristics to boundary forcing perturbations. *Climate of the Past*, 19(1):179–198, Jan. 2023. ISSN 1814-9332. doi: 10.5194/cp-19-179-2023. URL <http://dx.doi.org/10.5194/cp-19-179-2023>.

- G. Vettoretti and W. R. Peltier. Fast physics and slow physics in the nonlinear dansgaard–oeschger relaxation oscillation. *Journal of Climate*, 31(9):3423–3449, Mar. 2018. ISSN 1520-0442. doi: 10.1175/jcli-d-17-0559.1. URL <http://dx.doi.org/10.1175/JCLI-D-17-0559.1>.
- G. Vettoretti, P. Ditlevsen, M. Jochum, and S. O. Rasmussen. Atmospheric co2 control of spontaneous millennial-scale ice age climate oscillations. *Nature Geoscience*, 15(4):300–306, Apr. 2022. ISSN 1752-0908. doi: 10.1038/s41561-022-00920-7. URL <http://dx.doi.org/10.1038/s41561-022-00920-7>.

Reviewer #2 (Remarks to the Author):

The manuscript on "A mechanism for reconciling the synchronisation of Heinrich events and Dansgaard-Oeschger cycles" has been substantially revised in response to prior reviews. In general, I think the manuscript has substantially improved and that the authors have been responsive to my prior suggestions and those of the other reviewers, particularly in clarity of the explanation, the claims being made and the relationship of this study to prior literature.

In this review, I can hone my critique to one critical aspect of the discussion that I do not feel has been resolved by the revision. In particular, this related to my previous review point (P2.5 in the response to reviewers document) on explaining the role of surface temperature perturbations in the phase locking mechanism described in the paper. Since this is the central aspect of this study which is novel (the atmospheric mechanism that is), I think it is quite important that it be clear that surface temperature perturbations do indeed play a role in the phase locking. I have two outstanding questions:

1. It is not clear to me that the surface temperature perturbations are strictly necessary (separate from the SMB perturbations) to achieve phase locking. The revised text says the following (my asterisk emphasis added): "The warmer surface temperatures lead to increased ice deformation and enhanced frictional heat generation, while the higher surface mass balance flux increases ice stream thickening. *Together* these processes ensure that a critical ice thickness and basal temperature threshold, required for the onset of the internal instability, is reached towards the end of the interstadial phase". Reading this and looking at figure 2, what I cannot tell is whether the surface temperature perturbation is strictly necessary to produce phase locking. What would be helpful to show is whether phase locking occurs in experiments with zero surface temperature perturbations and non-zero SMB perturbation, or conversely in experiments with non-zero surface temperature perturbations and zero SMB perturbation.

2. Part of the reason for my raising the issue above, is that I am still having a hard time understanding the mechanism proposed for surface temperature perturbation influence on basal sliding on time scales of less than 1000 years. The response says "the driving process here is that the surface warming results in an enhanced ice deformation because the ice deformation law is dependent on ice temperature. This in turn raises the internal frictional heat generation and slightly warms the ice temperature at the ice base." Where I have the disconnect is that the temperature effect and effect of ice deformation and internal heat generation should be confined to the upper 100 or so meters on time scales of less than 1000 years (see MacAyeal 1993 in Paleoceanography for this argument explicitly). On that same time scale, the change in temperature at the bed is likely to be minuscule (I'd estimate order 10^{-11} C). Thus, even with feedbacks associated with changes in ice flow and internal heat generation, those effects would still be confined to the upper part of the ice sheet. Given how cold the ice is near the surface, the surface temperature perturbations aren't likely to produce internal melt (which could conceivably drain to the base), and so I remain at a loss for how the temperature signal (absent SMB forcing) is meant to propagate to the base of the ice sheet in less than 1000 years. It is possible that in performing the vertical integration to solve the momentum balance with the hybrid scheme described here, the change in temperature at the surface effectively gets included in the basal sliding momentum balance. However, if that is the case, then this effect is an artifact of the model formulation, not a physical mechanism for propagating surface temperature perturbations to the base.

Other than these issues, I am happy with the manuscript and where it is, but given that the outstanding issues related to the central mechanism novel to this paper, it bears ensuring that it is (a) well described, (b) shown convincingly to be occurring using careful experiments, and (c) due to real physical processes and not model formulation.

Reviewer #3 (Remarks to the Author):

The Authors have addressed my comments and have done an excellent job in revising the manuscript. I have no further comments.

Response to the reviewers

We thank all referees for their thoughtful and thorough reviews of our paper. We appreciate you taking the time to complete these reviews and welcome your helpful comments. In the following we address their concerns point by point. Throughout this response to review document referee comments are provided in regular, non- italic font text, our response comments are provided in red font.

To accomodate the main comment of the reviewer, we have made the following main changes to the manuscript:

- We included results from six additional sensitivity simulations which show, if the forcing is large enough, one of the two considered atmospheric perturbations is sufficient to achieve phase locking.

Reviewer 1

Reviewer Point P 1.1 — The manuscript on "A mechanism for reconciling the synchronisation of Heinrich events and Dansgaard-Oeschger cycles" has been substantially revised in response to prior reviews. In general, I think the manuscript has substantially improved and that the authors have been responsive to my prior suggestions and those of the other reviewers, particularly in clarity of the explanation, the claims being made and the relationship of this study to prior literature.

In this review, I can hone my critique to one critical aspect of the discussion that I do not feel has been resolved by the revision. In particular, this related to my previous review point (P2.5 in the response to reviewers document) on explaining the role of surface temperature perturbations in the phase locking mechanism described in the paper. Since this is the central aspect of this study which is novel (the atmospheric mechanism that is), I think it is quite important that it be clear that surface temperature perturbations do indeed play a role in the phase locking. I have two outstanding questions:

Reply: We thank the reviewer for the positive assessment of our revised manuscript. We address the remaining concerns in detail below.

Reviewer Point P 1.2 — It is not clear to me that the surface temperature perturbations are strictly necessary (separate from the SMB perturbations) to achieve phase locking. The revised text says the following (my asterisk emphasis added): "The warmer surface temperatures lead to increased ice deformation and enhanced frictional heat generation, while the higher surface mass balance flux increases ice stream thickening. *Together* these processes ensure that a critical ice thickness and basal temperature threshold, required for the onset of the internal instability, is reached towards the end of the interstadial phase". Reading this and looking at figure 2, what I cannot tell is whether the surface temperature perturbation is strictly necessary to produce phase locking. What would be helpful to show is whether phase locking occurs in experiments with zero surface temperature perturbations and non-zero SMB perturbation, or conversely in experiments with non-zero surface temperature perturbations and zero SMB perturbation.

Reply: We agree with the reviewer that these type of simulations would be helpful. So, we have performed in total six new simulations. In three of these simulations, only the surface mass balance was perturbed and in the other three simulations only surface temperature was perturbed. The results show that if the forcing is strong enough, one of the two forcings is sufficient to achieve phase locking. We have added the corresponding figures to the extended figures section and added the following text to the manuscript: “Additional sensitivity simulations reveal that for the stronger perturbation scenarios ($\Delta T_{emp} \geq 10^{\circ}\text{C}$ and $\Delta SMB \geq 0.075 \text{ m/yr}$) only one of the two atmospheric forcings is required to achieve phase locking. Hereby, perturbations to only the surface mass balance result in Heinrich events occurring deeper into the stadial phase of the Dansgaard-Oeschger cycle.”

Reviewer Point P 1.3 — Part of the reason for my raising the issue above, is that I am still having a hard time understanding the mechanism proposed for surface temperature perturbation influence on basal sliding on time scales of less than 1000 years. The response says “the driving process here is that the surface warming results in an enhanced ice deformation because the ice deformation law is dependent on ice temperature. This in turn raises the internal frictional heat generation and slightly warms the ice temperature at the ice base.” Where I have the disconnect is that the temperature effect and effect of ice deformation and internal heat generation should be confined to the upper 100 or so meters on time scales of less than 1000 years (see MacAyeal 1993 in Paleooceanography for this argument explicitly). On that same time scale, the change in temperature at the bed is likely to be minuscule (I’d estimate order 10^{-11}°C). Thus, even with feedbacks associated with changes in ice flow and internal heat generation, those effects would still be confined to the upper part of the ice sheet. Given how cold the ice is near the surface, the surface temperature perturbations aren’t likely to produce internal melt (which could conceivably drain to the base), and so I remain at a loss for how the temperature signal (absent SMB forcing) is meant to propagate to the base of the ice sheet in less than 1000 years. It is possible that in performing the vertical integration to solve the momentum balance with the hybrid scheme described here, the change in temperature at the surface effectively gets included in the basal sliding momentum balance. However, if that is the case, then this effect is an artifact of the model formulation, not a physical mechanism for propagating surface temperature perturbations to the base

Reply: We agree with the reviewer that the effect of the surface temperature on the basal temperature is not massive on the 1,500 year cycle considered here. However, there is an important difference between MacAyeal’s work and the work we present here. MacAyeal assumes a spatially homogenous ice thickness of 3,000 meters for Hudson ice stream. While our ice stream shows a similar ice thickness near the drainage divide, the Heinrich events are initiated close to the ice sheet margin and propagate upstream. In this region, ice thickness is around 700 m prior to a Heinrich event in our model simulations. While the reviewer is correct that the e-folding depth, assuming no vertical advection, is 75 m and 145 m for 400 year and 1,500 year cycle respectively, the temperature warming at the base of the region still amounts to 0.002°C and 0.1°C at 700 m depth for the largest of our temperature perturbation (14°C), respectively. Naturally, these numbers would only increase if vertical advection and frictional heat generation would also be considered. Overall, this certainly presents not a massive warming, but because the base of the ice sheet is already very close to the pressure-melting-point, it is enough to nudge a few cells across the pressure-melting-point threshold, resulting in a different timing of the Heinrich event. However, as shown in our additional sensitivity simulations, this effect only comes into play for sufficiently large enough surface temperature perturbations.

Reviewer Point P 1.4 — Other than these issues, I am happy with the manuscript and where it is, but given that the outstanding issues related to the central mechanism novel to this paper, it bears ensuring that it is (a) well described, (b) shown convincingly to be occurring using careful experiments, and (c) due to real physical processes and not model formulation.

Reply: We hope that our above explanations and additions to the manuscript address the concerns of the reviewer.

Reviewer 2

Reviewer Point P 2.1 — The Authors have addressed my comments and have done an excellent job in revising the manuscript. I have no further comments.

Reply: We are happy that the reviewer is satisfied with our revisions to the initial submission.

Reviewer #2 (Remarks to the Author):

I am satisfied by the new simulations and comments made by the reviewers addressing my concerns.
I have no further suggestions.

Reviewer 1

Reviewer Point P 1.1 — I am satisfied by the new simulations and comments made by the reviewers addressing my concerns. I have no further suggestions.

Reply: We are happy that the reviewer is satisfied with our final revisions to the previous submission.